# Application of Referenced Thermodynamic Integration to Bayesian Model Selection

## Abstract

Evaluating normalising constants is important across a range of topics in statistical learning, notably Bayesian model selection. However, in many realistic problems this involves the integration of analytically intractable, high-dimensional distributions, and therefore requires the use of stochastic methods such as thermodynamic integration (TI). In this paper we apply a simple but under-appreciated variation of the TI method, here referred to as *referenced TI*, which computes a single model's normalising constant in an efficient way by using a judiciously chosen reference density. The advantages of the approach and theoretical considerations are set out, along with pedagogical 1 and 2D examples. The approach is shown to be useful in practice when applied to a real problem — to perform model selection for a semi-mechanistic hierarchical Bayesian model of COVID-19 transmission in South Korea involving the integration of a 200D density.

## 1 Introduction

The marginalised likelihood, or normalising constant of a model, is a feature central to the principles and pathology of Bayesian statistics. For example — given two models representing two competing hypotheses, the ratio of the normalising constants (known as the Bayes factor), describes the relative probability of the data having been generated by one hypothesis compared to the other. Consequently, at a practical level the estimation of normalising constants is an important topic for model selection in the Bayesian setting (Kass & Raftery, 1995).

In practice, estimating a normalising constant relies on 'integrating out' or marginalising the parameters of the model to get the probability the associated hypothesis produced the data. But in general this is difficult, because we cannot easily integrate arbitrary high-dimensional distributions — certainly analytical or quadrature-based methods are of little help directly. As a result, practitioners turn to a range of approaches, typically based on statistical sampling. Specific examples include bridge sampling (Bennett, 1976; Meng & Wong, 1996), stochastic density of states based methods (Skilling, 2006; Habeck, 2012) and thermodynamic integration (Kirkwood, 1935; Gelman & Meng, 1998; Lartillot & Philippe, 2006; Friel et al., 2017). This work focuses on the latter — thermodynamic integration — in particular on the development of the practical details for efficient application for Bayesian model selection.

By way of introduction for researchers unfamiliar with thermodynamic integration (TI), it allows us to estimate the ratio of two normalising constants in a general and asymptotically exact way. Instead of marginalising the associated densities explicitly in terms of the high-dimensional integrals, using TI we only have to evaluate a 1-dimensional integral, where the integrand can easily be sampled with Markov Chain Monte Carlo (MCMC). To see how this works, consider two models labelled 1 and 2 with normalising constants $z_1$ and $z_2$. Each $z$ is given by

$$z_i = \int q_i(\boldsymbol{\theta})d\boldsymbol{\theta}, \ i \in \{1,2\}, \tag{1}$$

where $q_i$ is a density for model $M_i$ with parameters $\boldsymbol{\theta}$, that gives the model's Bayesian posterior density as

$$p_i(\boldsymbol{\theta}) = \frac{q_i(\boldsymbol{\theta})}{z_i}, \ i \in \{1,2\}.$$

To apply thermodynamic integration we introduce the concept of a path between $q_1(\boldsymbol{\theta})$ and $q_2(\boldsymbol{\theta})$, linking the two densities via a series of intermediate ones. This family of densities, parameterised by the path coordinate $\lambda$, is denoted by $q(\lambda; \boldsymbol{\theta})$. An example path in $\lambda$ is shown in Fig. 1A.

The density $q(\lambda; \boldsymbol{\theta})$, linking $q_1$ to $q_2$ and defining the intermediate densities, can be chosen to have an optimal or in some way convenient path. A common choice based on convenience is the geometric one

$$q(\lambda; \boldsymbol{\theta}) = q_2^{\lambda}(\boldsymbol{\theta}) q_1^{1-\lambda}(\boldsymbol{\theta}), \ \lambda \in [0, 1].$$

The important point to note is that for $\lambda = 0$, $q(\lambda; \boldsymbol{\theta})$ returns the first density $q(0; \boldsymbol{\theta}) = q_1(\boldsymbol{\theta})$, for $\lambda = 1$ it gives $q(1; \boldsymbol{\theta}) = q_2(\boldsymbol{\theta})$, and for in-between $\lambda$ values a log-linear mixture of the endpoint densities. Just as we have defined a family of densities, there is an associated normalising constant for any point along the path, that for any value of $\lambda$ is given by

$$z(\lambda) = \int_{\Omega(\lambda)} q(\lambda; \boldsymbol{\theta}) d\boldsymbol{\theta}.$$

A further small but important point to avoid complications is to have densities with common support, for example $\Omega(\lambda = 1) = \Omega(\lambda = 0)$. Hereafter support is denoted by $\Omega$.

Having set up the definitions of $q(\lambda; \boldsymbol{\theta})$ and $z(\lambda)$, the TI expression can be derived, to compute the log-ratio of $z_1 = z(\lambda = 0)$ and $z_2 = z(\lambda = 1)$, while avoiding explicit integrals over the models' parameters $\boldsymbol{\theta}$. This is laid out as follows:

$$
\begin{aligned}
\log \frac{z_2}{z_1} &= \int_0^1 \partial_\lambda \log z(\lambda) \, d\lambda \\
&= \int_0^1 \frac{1}{z(\lambda)} \partial_\lambda z(\lambda) \, d\lambda \\
&= \int_0^1 \frac{1}{z(\lambda)} \partial_\lambda \int_\Omega q(\lambda; \boldsymbol{\theta}) \, d\boldsymbol{\theta} \, d\lambda \\
&= \int_0^1 \frac{1}{z(\lambda)} \int_\Omega \left( \partial_\lambda \log q(\lambda; \boldsymbol{\theta}) \right) q(\lambda; \boldsymbol{\theta}) \, d\boldsymbol{\theta} \, d\lambda \\
&= \int_0^1 \mathbb{E}_{p(\lambda; \boldsymbol{\theta})} \left[ \partial_\lambda \log q(\lambda; \boldsymbol{\theta}) \right] \, d\lambda \\
&= \int_0^1 \mathbb{E}_{p(\lambda; \boldsymbol{\theta})} \left[ \log \frac{q_2(\boldsymbol{\theta})}{q_1(\boldsymbol{\theta})} \right] \, d\lambda \\
&= \int_0^1 \mathbb{E}_{q(\lambda; \boldsymbol{\theta})} \left[ \log \frac{q_2(\boldsymbol{\theta})}{q_1(\boldsymbol{\theta})} \right] \, d\lambda,
\end{aligned}
\tag{2}
$$

Here we started with the fundamental theorem of calculus (first step), rules of differentiating logs (second step), definition of $z(\lambda)$ (third step), assumed exchangeability of $\partial_\lambda$ and $\int d\boldsymbol{\theta}$ and log differentiation rules again (fourth step), identifying the expectation $\mathbb{E}_{q(\lambda; \theta)}$ from sampling distribution $q(\lambda; \theta)$ (fifth step), differentiation of the geometric path for $q(\lambda)$ (sixth step), and finally equivalence of sampling from $q$ and $p$. The final line in the expression summarises the usefulness of TI. Instead of having to work with the complicated high-dimensional integrals of Equation 1 to find the log-Bayes factor $\log \frac{z_2}{z_1}$, which measures the relative probability of getting the data from one hypothesis compared to another, we only need to consider a 1-dimensional integral of an expectation, and that expectation can be readily produced by MCMC.

In our paper we examine the details of a *referenced TI* approach, which is a variation on the TI theme that we find useful to enable fast and accurate normalising constant calculations. Our main contributions are as follows:

- We show how to generate a reference normalising constant from an exactly-integratable reference density, through sampling or gradients, and with parameter constraints. And we present how to use this reference in the TI method to efficiently estimate a normalising constant of an arbitrary high-dimensional density.

- We discuss performance benchmarks for a well-known problem in the statistical literature (Williams, 1959), which shows the method performs favourably in terms of accuracy and the number of iterations to convergence.

- Finally the technique is applied to a hierarchical Bayesian time-series model describing the COVID-19 epidemic in South Korea.

In relation to other work, we recognise using a reference for thermodynamic integration is a topic that has been raised, especially in early theoretically-oriented literature (Neal, 1993; Diciccio et al., 1997; Grosse et al., 2013). Our additional contribution is to bridge the gap from theory and simple examples to application, which includes choosing the reference using MCMC samples or gradients, examination of reference support, comparisons of convergence, and illustration of the approach for a non-trivial real-world problem.

## 2 Referenced-TI

Introducing a reference density and associated normalising constant as

$$
\begin{aligned}
z & = z_{\text{ref}} \frac{z}{z_{\text{ref}}} \\
& = z_{\text{ref}} \exp \int_0^1 \mathbb{E}_{q(\lambda;\boldsymbol{\theta})} \left[ \log \frac{q(\boldsymbol{\theta})}{q_{\text{ref}}(\boldsymbol{\theta})} \right] d\lambda,
\end{aligned}
\tag{3}
$$

yields an efficient approach to compute Bayes factors, or more generally to marginalise an arbitrary density for any application. To clarify notation, here $z$ is the normalising constant of interest with density $q$, $z_{\text{ref}}$ is a reference normalising constant with associated density $q_{\text{ref}}$. In the second line the ratio $z/z_{\text{ref}}$ is straightforwardly given by the thermodynamic integral identity in Equation 2.

While the Equation 2 can be directly applied to conduct a pairwise model comparison between two hypotheses, by introducing a reference we can naturally marginalise the density of a single model (Neal, 1993; Diciccio et al., 1997). This is useful when comparing multiple models as $n > \binom{n}{2}$ for $n > 3$. Another motivation to reference the TI is the MCMC computational efficiency of converging the TI expectation. In Equation 3, with judicious choice of $q_{\text{ref}}$, the reference normalising constant $z_{\text{ref}}$ can be evaluated analytically and account for most of $z$. In this case $\log \frac{q(\boldsymbol{\theta})}{q_{\text{ref}}(\boldsymbol{\theta})}$ tends to have a small expectation and variance and converges quickly.

This idea of using an exactly solvable reference, to aid in the solution of an otherwise intractable problem, has been a recurrent theme in the computational and mathematical sciences in general (Dirac, 1927; Gell-Mann & Brueckner, 1957; Vocadlo & Alfe, 2002; Duff et al., 2015), and variations on this approach have been used to compute normalising constants in various guises in the statistical literature (Friel et al., 2017; Lartillot & Philippe, 2006; Xie et al., 2010; Neal, 1993; Cameron & Pettitt, 2014; Friel & Pettitt, 2008; Friel & Wyse, 2012; Lefebvre et al., 2010; Fan et al., 2011; Baele et al., 2016). For example, in the generalised stepping stone method a reference is introduced to speed up convergence of the importance sampling at each temperature rung (Fan et al., 2011; Baele et al., 2016). In the work of Lefebvre et al. (2010) a theoretical discussion has been presented that shows the error budget of thermodynamic integration depends on the J-divergence of the densities being marginalised. Noting this, Cameron & Pettitt (2014) provide an illustration for a 2-dimensional example in their work on recursive pathways to marginal likelihood estimation. And in the power posteriors method, a reference is used but the reference is a prior density and thus $z_{\text{ref}} = 1$ (Friel & Pettitt, 2008). This approach is elegant as the reference need not be chosen — it is simply the prior — however the downside is that for poorly chosen or uninformative priors, the thermodynamic integral will be slow to converge and susceptible to instability. In particular for complex hierarchical models with weakly informative priors this is found to be an issue.

For referenced TI as presented here, the reference density in Equation 3 can be chosen at convenience, but the main desirable features are that it should be easily formed without special consideration or adjustments and that $z_{\text{ref}}$ should be analytically integratable and account for as much of $z$ as possible. Such a choice of $z_{\text{ref}}$ ensures the part with expensive sampling is small and converges quickly. An obvious choice in this

regard is the Laplace-type reference, where the log-density is approximated with a second-order one, for example a multivariate Gaussian. For densities with a single concentration, Laplace-type approximations are ubiquitous, and an excellent natural choice for many problems. In the following section we consider approaches that can be used to formulate a reference normalising constant $z_{\text{ref}}$ from a second-order log-density (though more generally other tractable references are possible). In each referenced TI scenario, we note that even if the reference approximation is poor, the estimate of the normalising constant based on Equation 3 remains asymptotically exact—only the speed of convergence is affected (subject to the assumptions of matching support for end-point densities).

## 2.1 Taylor Expansion at the Mode Laplace Reference

The most straightforward way to generate a reference density is to Taylor expand the log-density to second order about a mode. Noting no linear term is present, we see the reference density is

$$q_{\text{ref}}(\boldsymbol{\theta}) \;=\; \exp\left(\log q(\boldsymbol{\theta}_0) + \frac{1}{2}(\boldsymbol{\theta} - \boldsymbol{\theta}_0)^{\text{T}} \mathbf{H}\,(\boldsymbol{\theta} - \boldsymbol{\theta}_0)\right), \tag{4}$$

where $\mathbf{H}$ is the Hessian matrix and $\boldsymbol{\theta}_0$ is the vector of mode parameters. The associated normalising constant is

$$\begin{aligned}
z_{\text{ref}} \;&=\; \int_{-\infty}^{\infty} q_{\text{ref}}(\boldsymbol{\theta}) d\boldsymbol{\theta} \\
&=\; \int_{-\infty}^{\infty} \exp\left(\log q(\boldsymbol{\theta}_0) + \frac{1}{2}(\boldsymbol{\theta} - \boldsymbol{\theta}_0)^{\text{T}} \mathbf{H}\,(\boldsymbol{\theta} - \boldsymbol{\theta}_0)\right) d\boldsymbol{\theta} \\
&=\; q(\boldsymbol{\theta}_0) \int_{-\infty}^{\infty} \exp\left(\frac{1}{2}(\boldsymbol{\theta} - \boldsymbol{\theta}_0)^{\text{T}} \mathbf{H}\,(\boldsymbol{\theta} - \boldsymbol{\theta}_0)\right) d\boldsymbol{\theta} \\
&=\; q(\boldsymbol{\theta}_0) \sqrt{\det(2\pi \mathbf{H}^{-1})}.
\end{aligned} \tag{5}$$

This approach to yield a reference density, using either analytic or finite difference gradients at mode, tends to produce a density close to the true one in the neighbourhood of $\boldsymbol{\theta}_0$. But this is far from guaranteed, particularly if the density is asymmetric, or has non-negligible high-order moments, or is discontinuous for example exhibiting cusps. In many instances a more reliable choice of reference can be found by using MCMC samples from the whole posterior density.

## 2.2 Sampled Covariance Laplace Reference

A second straightforward approach to form a reference density, that's often more robust, is by drawing samples from the true density $q(\boldsymbol{\theta})$ to estimate the mean parameters $\hat{\boldsymbol{\theta}}$ and covariance matrix $\hat{\boldsymbol{\Sigma}}$, such that

$$q_{\text{ref}}(\boldsymbol{\theta}) \;=\; q(\hat{\boldsymbol{\theta}}) \exp\left(-\frac{1}{2}(\boldsymbol{\theta} - \hat{\boldsymbol{\theta}})^{\text{T}} \hat{\boldsymbol{\Sigma}}^{-1}\,(\boldsymbol{\theta} - \hat{\boldsymbol{\theta}})\right) \tag{6}$$

Then the reference normalising constant is

$$z_{\text{ref}} = q(\hat{\boldsymbol{\theta}}) \sqrt{\det(2\pi \hat{\boldsymbol{\Sigma}})}. \tag{7}$$

This method of generating a reference is simple and reliable. It requires sampling from the posterior $q(\boldsymbol{\theta})$ so is more expensive than gradient-based methods, but the cost associated with drawing enough samples to generate a sufficiently good reference tends to be quite low. In the primary application discussed later, regarding structured high-dimensional Bayesian hierarchical models, we use this approach to generate a reference density and normalising constant.

Though the sampled covariance reference is typically a good approach, it is not in general optimal within the Laplace-type family of approaches — typically another Gaussian reference exists with different parameters that can generate a normalising constant closer to the true one, thus potentially leading to overall faster convergence of the thermodynamic integral to the exact value. Such an optimal reference can be identified variationally, as we show in Section A.1.

### 2.3 Reference Support

If a model involves a bounded parameter space, for example $\theta_1 \in [0, \infty)$, $\theta_2 \in (-1, \infty)$ etc. as commonly arise in structured Bayesian models, in referenced TI the exact analytic integration for the reference density should be commensurately limited. This is necessary not only so the reference is closer to the true density to speed up convergence, but also so MCMC samples from both densities can be drawn on the same parameter space, as is required for the thermodynamic integrand in Equation 3 to be well-defined. However, the calculation of arbitrary probability density function orthants, even for well-known analytic functions such as the multivariate Gaussian, is in general a difficult problem. High-dimensional orthant computations usually require advanced techniques, the use of approximations, or sampling methods (Ridgway, 2016; Azzimonti & Ginsbourger, 2018; Owen, 2014; Miwa et al., 2003; Curnow & Dunnett, 1962; Ruben, 1964). Fortunately, we can simplify our reference density to create a reference with tractable analytic integration for limits by using a diagonal approximation to the sampled covariance or Hessian matrix. For example the orthant of a diagonal multivariate Gaussian can be given in terms of the error function (Brown, 1963),

$$z_{\text{ref}} = q(\hat{\boldsymbol{\theta}})\sqrt{\det(2\pi\boldsymbol{\Sigma}^{\text{diag}})} \prod_{i \in K}\left(1 + \text{erf}\left(\frac{\hat{\theta}_i - a_i}{\sqrt{2\Sigma_i^{\text{diag}}}}\right)\right), \tag{8}$$

where $K$ denotes the set of indices of the parameters with lower limits $a_i$. $\Sigma^{\text{diag}}$ is a diagonal covariance matrix, that is one containing only the variance of each of the parameters, without the covariance terms and $\Sigma_i^{\text{diag}}$ denotes the $i^{\text{th}}$ element of the diagonal. Restricting our density to a diagonal one is a poorer approximation than using the full covariance matrix. In practice however this has not been a substantial drawback to the convergence of the thermodynamic integral—and again we state that the quality of the reference affects only convergence rather than eventual accuracy of the normalising constant. This behaviour is observed in the practical examples later considered, though we do recognise the distinction between accuracy and convergence and matters of asymptotic consistency using an MCMC estimator with finite iterations are not clear cut.

### 2.4 Technical Implementation

Referenced TI was implemented in Python and Stan programming languages. Using Stan enables fast MCMC simulations with Hamiltonian Monte Carlo and No-U-Turn algorithm (Hoffman & Gelman, 2014; Carpenter et al., 2017), and portability between other statistical languages. We also provide an example of carrying out referenced-TI in NumPyro (Phan et al., 2019). The code for all examples shown in this paper is available at https://github.com/revealed-when-accepted. In the examples shown in Section 3, we used 4 chains with 20,000 iterations per chain for the pedagogical examples, and 4 chains with 2,000 iterations for the other applications. In all cases, half of the iterations were used for the warm-up. Mixing of the chains and the sampling convergence was checked in each case, by ensuring that the $\hat{R}$ value was $\leq 1.05$ and investigating the trace plots.

In all examples, the integral given in Equation 2 was discretised to allow computer simulations. Each expectation $\mathbb{E}_{q(\lambda;\theta)}\left[\log \frac{q_1(\theta)}{q_0(\theta)}\right]$ was evaluated at $\lambda = 0.0, 0.1, 0.2, ...., 0.9, 1.0$, unless stated otherwise. To obtain the value of the integral in Equation 2, we interpolated a curve linking the expectations using a cubic spline, which was then integrated numerically. The pseudo-code of the algorithm with sampled covariance Laplace reference is shown in Section A.2.

## 3 Applications

In this section we present applications of the referenced TI approach. In 3.1 and 3.2 we give 1- and 2-dimensional pedagogical introductions to the approach. In 3.3 we select a linear regression model for a well-known problem in the statistical literature, and finally in 3.4 we consider a challenging model selection task for a structured Bayesian model of the COVID-19 epidemic in South Korea.

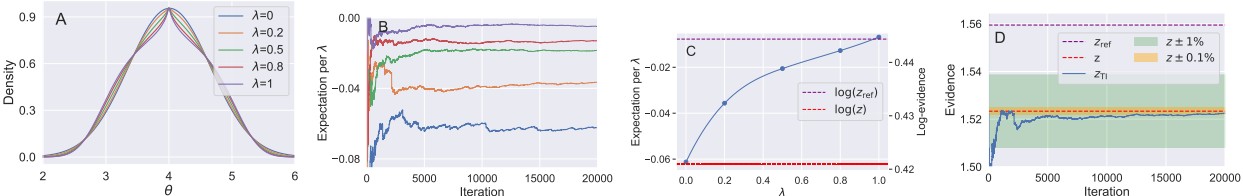

Figure 1: A) $q^\lambda q_{\text{ref}}^{(1-\lambda)}$ for the 1d example density in parameter $\theta$ (Equation 9) at selected $\lambda$ values along the path. B) Expectation $\mathbb{E}_{q(\lambda;\theta)}\left[\log\frac{q(\theta)}{q_{\text{ref}}(\theta)}\right]$ vs MCMC iteration, shown at each value of $\lambda$ sampled. C) $\lambda$-dependence of $\mathbb{E}_{q(\lambda;\theta)}\left[\log\frac{q(\theta)}{q_{\text{ref}}(\theta)}\right]$, the TI contribution to the log-evidence. D) Convergence of the evidence $z$, with 1% convergence after 500 iterations and 0.1% after 17,000 iterations per $\lambda$.

### 3.1 1D Pedagogical Example

To illustrate the technique we consider the 1-dimensional density

$$q(\theta) = \exp\left(-\frac{1}{2}\sqrt{|\theta - 4|} - \frac{1}{2}(\theta - 4)^4\right), \ \theta \in \mathbb{R}, \tag{9}$$

with normalising constant $z = \int_{-\infty}^{\infty} q(\theta)d\theta$. This density has a cusp and it does not have an analytical integral that easily generalises to multiple dimensions.

In this instance the Laplace approximation based on the second-order Taylor expansion at the mode will fail due to the cusp, so we use the more robust covariance sampling method. Sampling from the 1D density $q(\theta)$ we find a variance of $\hat{\sigma}^2 = 0.424$, giving a Gaussian reference density $q_{\text{ref}}(\theta)$ with normalising constant of $z_{\text{ref}} = 1.559$. The full normalising constant, $z = z_{\text{ref}}\frac{z}{z_{\text{ref}}}$, is evaluated by Equation 3, by setting up a thermodynamic integration along the sampling path $q^\lambda q_{\text{ref}}^{(1-\lambda)}$. The expectation, $\mathbb{E}_{q(\lambda;\theta)}\left[\log\frac{q(\theta)}{q_{\text{ref}}(\theta)}\right]$, is evaluated at 5 points along the coupling parameter path $\lambda = 0.0, 0.2, 0.5, 0.8, 1.0$, shown in Fig. 1. In this simple example, the integral can be easily evaluated to high accuracy using quadrature (Piessens et al., 1983; Virtanen et al., 2020), giving a value of 1.523. Referenced TI reproduces this value, with convergence of $z$ shown in Fig. 1, converging to 1% of $z$ with 500 iterations and 0.1% within 17,000 iterations.

This example illustrates notable characteristic features of referenced TI. Here the reference $q_{\text{ref}}(\theta)$ is a good approximation to $q(\theta)$, with $z_{\text{ref}}$ accounting for most of $z$ ($z_{\text{ref}} = 1.02z$). Consequently $\frac{z}{z_{\text{ref}}}$ is close to 1, and the expectations, $\mathbb{E}_{q(\lambda;\theta)}\left[\log\frac{q(\theta)}{q_{\text{ref}}(\theta)}\right]$, evaluated by MCMC for the remaining part of the integral are small. For the same reasons the variance at each $\lambda$ is small, leading to favourable convergence within a small number of iterations. And finally $\mathbb{E}_{q(\lambda;\theta)}\left[\log\frac{q(\theta)}{q_{\text{ref}}(\theta)}\right]$ weakly depends on $\lambda$, so there is no need to use a very fine grid of $\lambda$ values or consider optimal paths—satisfactory convergence is easily achieved using a simple geometric path with 4 $\lambda$-intervals.

### 3.2 2D Pedagogical Example with Constrained Parameters

As a second example, consider a 2-dimensional unnormalised density function with a constrained parameter space:

$$q(\theta_1, \theta_2) = \exp(-\Theta), \tag{10}$$

with

$$\Theta = \frac{1}{4}\sum_{i,j\in\{1,2\}}\left(\theta_i + \frac{1}{2}\right)^{2j} + \frac{1}{8}\theta_1\theta_2^2, \tag{11}$$

and

$$\theta_1 \in [0, +\infty) \text{ and } \theta_2 \in (-\infty, +\infty). $$

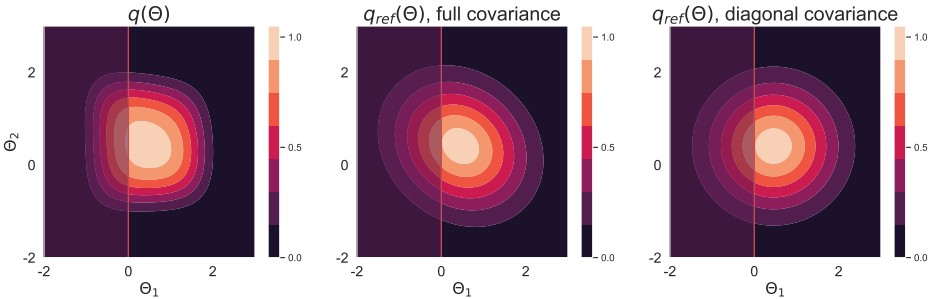

Figure 2: Contour plots of the un-normalised density $q$ and its two reference densities $q_{\text{ref}}$, one using a full covariance matrix and another using a diagonal covariance matrix that can be easily marginalised. The red line shows the lower boundary $\theta_1 = 0$ and the shaded $\theta_1 < 0$ region to the left of the line is outside of the support of the density $q$.

A reference density $q_{\text{ref}}(\boldsymbol{\theta})$ can be constructed from the Hessian at the mode of $q(\boldsymbol{\theta})$. To marginalise it on a parameter space with restricted support, we use a reference density $q_{\text{ref}}^{\text{diag}}(\boldsymbol{\theta})$ based on a diagonal Hessian, that has an exact and easy to calculate orthant. All densities are shown in Fig. 2.

To obtain the log-evidence of the model, we calculated the exact value numerically (Piessens et al., 1983; Virtanen et al., 2020), and using a sampled diagonal covariance matrix, as per Equation 8, to account for the lower bound of the parameter $\theta_1$. Without this restriction the final normalising constant is overestimated – if the support of the parameters in the MCMC isn't the same as for the analytic $z_{\text{ref}}$ calculation, $z_{\text{ref}}$ as shown in Equation 3 doesn't cancel with the TI reference. Numerical comparison of the referenced TI to quadrature is presented in Table 2 in the Appendix.

### 3.3 Benchmarks—*Radiata Pine*

To benchmark the application of the referenced TI in the model selection task, two non-nested linear regression models are compared for the *radiata pine* data set (Williams, 1959). This example has been widely used for testing normalising constant calculating methods, since in this instance the exact value of the model evidence can be computed. The data consists of 42 3-dimensional data-points, expressed as $y_i$ - maximum compression strength, $x_i$ - density and $z_i$ - density adjusted for resin content. In this example, we follow the approach of Friel & Wyse (2012), using the priors from therein, and test which of the two models $M_1$ and $M_2$ provides better predictions for the compression strength:

$$M_1 : y_i = \alpha + \beta(x_i - \bar{x}) + \epsilon_i, \epsilon_i \sim N(0, \tau^{-1}), i = 1, ..., n \,,$$

$$M_2 : y_i = \gamma + \delta(z_i - \bar{z}) + \eta_i, \eta_i \sim N(0, \rho^{-1}), i = 1, ..., n \,.$$

Five methods of estimating the model evidence were used in this example: Laplace approximation using a sampled covariance matrix, model switch TI along a path directly connecting the models (Lartillot & Philippe, 2006; Vitoratou & Ntzoufras, 2017), referenced TI, power posteriors with equidistant 11 $\lambda$-placements (labelled here as $\text{PP}_{11}$) and power posteriors with 100 $\lambda$-s ($\text{PP}_{100}$), following the example from Friel & Wyse (2012). For the model switch TI, referenced TI and $\text{PP}_{11}$ we used $\lambda \in \{0.0, 0.1, ..., 1.0\}$.

The expectation from MCMC sampling per each $\lambda$ for model switch TI, referenced TI, $\text{PP}_{11}$ and $\text{PP}_{100}$ and fitted cubic splines between the expectations are shown in Fig. 5 in the Appendix. Both reference and model switch TI methods eliminate the problem of divergence of expectation for $\lambda = 0$, which is observed with the power posteriors, where samples for $\lambda = 0$ come from the prior density function. And both reference and model switch have smaller residuals for splines in $\lambda$ fitted to $\mathbb{E}_{q(\lambda;\boldsymbol{\theta})}\left[\log \frac{q(\boldsymbol{\theta})}{q_{\text{ref}}(\boldsymbol{\theta})}\right]$ than power posteriors.

For each approach, the splines fitted to $\mathbb{E}_{q(\lambda;\boldsymbol{\theta})}\left[\log \frac{q(\boldsymbol{\theta})}{q_{\text{ref}}(\boldsymbol{\theta})}\right]$ were integrated to obtain the log-evidence for models $M_1$ and $M_2$, and the log-ratio of the two models' evidences for the model switch TI. The rolling means of the integral over 1500 iterations for referenced TI and $\text{PP}_{100}$ for $M_2$ are shown in Fig. 3A. We

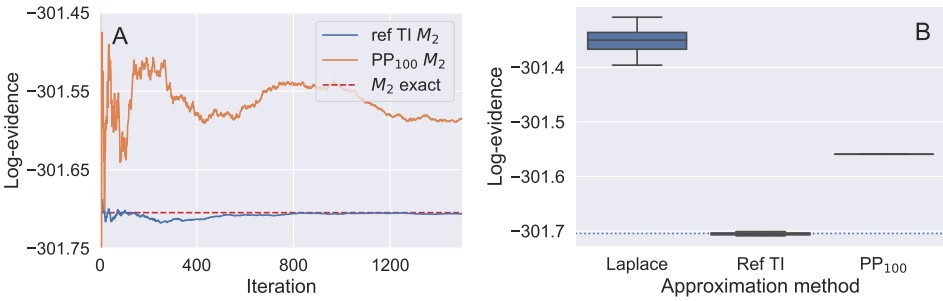

Figure 3: The log-evidence of $M_2$ from the Radiata Pine benchmark problem is shown estimated using three approaches. (A) shows the rolling mean of log-evidence of $M_2$ over 1500 iterations per $\lambda$ obtained by referenced TI (blue line) and $PP_{100}$ (orange line) methods. The exact value is shown with red dashed line. (B) shows the mean log-evidence of the model $M_2$ evaluated over 15 runs of the three algorithms. The exact value of the log-evidence is shown with the dotted line.

can see from the plot, that referenced TI presents favourable convergence to the exact value, whereas $PP_{100}$ oscillates around it. Fig. 3B shows the distribution of log-evidence for the same model generated by 15 runs of the three algorithms: Laplace approximation with sampled covariance matrix, referenced TI and $PP_{100}$. Similar figures for model $M_1$ are given in Fig. 6. Although all three methods resulted in a log-evidence satisfactorily close to the exact solution, referenced TI was the most accurate and importantly, converged fastest (308 MCMC draws compared to 55,000 draws needed for the power posterior method to achieve standard error of 0.5%, excluding warm-up, see Table 3 in the Appendix).

### 3.4 Model Selection for the COVID-19 Epidemic in South Korea

The final example of using referenced TI for calculating model evidence is fitting a renewal model to COVID-19 case data from South Korea. The data were obtained from open-data.ecdc.europa.eu/covid19/casedistribution/csv. The model is based on a statistical representation of a stochastic branching process whose expectation mechanistically follows a renewal-type equation. Its derivation and details are explained in Berah et al. (2021) and a short explanation of the model is provided in the Section A.3.3. Briefly, the model is fitted to the time-series case data and estimates a number of parameters, including serial interval and the effective reproduction number, $R_t$. The number of cases for each day are modelled by a negative binomial likelihood, with location parameter estimated by a renewal equation. Three modifications of the original model are tested here:

- variation of the infection generation interval for values $GI = 5, 6, 7, 8, 9, 10, 20$— where $GI$ denotes the mean of Rayleigh-distributed generation interval,

- changing the order of the autoregressive model for the reproduction number, for $AR(k)$ with $k = 2, 3, 4$ lags,

- varying the length of the sliding window for estimating the reproduction number for values in $W = 1, 2, 3, 4, 7$ days.

Within each group of models, $GI$, $AR$ and $W$, we want to select the best model through the highest evidence method. The dimension of each model was dependent on the modifications applied, but in all the cases the normalising constant was a 40- to 200-dimensional integral. The log-evidence of each model was calculated using the Laplace approximation with a sampled covariance matrix, and then correction to the estimate was obtained using referenced TI method. Values of the log-evidence for each model calculated by both Laplace and referenced TI methods are given in Table 1. Interestingly, the favoured model in each group, that is the model with the highest log-evidence, was different when the evidence was evaluated using the Laplace approximation than when it was evaluated with referenced TI. For example, using the Laplace method, sliding window of length 7 was incorrectly identified as the best model, whereas with referenced TI window

of length 2 was chosen to be the best among the tested sliding windows models, which agrees with the previous studies of the window-length selection in H1N1 influenza and SARS outbreaks (Parag & Donnelly, 2020). This exposes how essential it is to accurately determine the evidence, even good approximations can result in misleading results. Bayes factors for all model pairs are shown in Fig. 7 in the Appendix.

Table 1: Log-evidence estimated by Laplace and referenced TI approximations. In each section, model with the highest log-evidence estimated by Laplace or referenced TI method is indicated in bold. The credible intervals for log-evidence comes from calculating the quantiles of the integral from Equation 2, where the integral values were obtained from the spline interpolated using running means of the expecations per $\lambda$ over all iterations.

| Model | Log-evidence Laplace | Log-evidence ref TI [95% CrI] |
|---|---|---|
| GI=5 | -1274 | -716 [-715.6, -715.2] |
| GI=6 | -1274 | -703 [-703.3, -702.7] |
| GI=7 | -1255 | -732 [-732.4, -731.8] |
| GI=8 | -1245 | **-685** [-685.5, -684.7] |
| GI=9 | -1310 | -803 [-802.8, -802.3] |
| GI=10 | -1313 | -805 [-805.1, -805.3] |
| GI=20 | **-1170** | -796 [-796.3, -795.5] |
| AR(2) | **-1207** | -711 [-711.2, -710.6] |
| AR(3) | -1293 | **-704** [-704.7, -703.7] |
| AR(4) | -2166 | -821 [-820.6, -819.2.] |
| W=1 | -1260 | -802 [-802.1, -801.6] |
| W=2 | -1069 | **-791** [-791.2, -790.7] |
| W=3 | -1003 | -807 [-807.5, -807.2] |
| W=4 | -940 | -811 [-811.1, -810.7] |
| W=7 | **-875** | -814 [-813.7, -813.5] |

### 3.5 Interpretation of the COVID-19 model selection

The importance of performing model selection in a rigorous way is clear from Fig. 4, where the generated $R_t$ time-series are plotted for the models favoured by Laplace and referenced TI methods (additional posterior densities are shown in Fig. 8 in the Appendix). The differences in the $R_t$ time-series show the pitfalls of selecting an incorrect model. The differences between the two favoured models were most extreme for the $GI = 8$ and $GI = 20$ models. While a $GI = 8$ is plausible, even likely for COVID-19, $GI = 20$ is implausible given observed data (Bi et al., 2020). This is further supported by observing that for $GI = 20$, favoured by the Laplace method, $R_t$ reached the value of over 125 in the first peak—around 100 more than for the $GI = 8$. The second peak was also largely overestimated, where $R_t$ reached a value of 75.

We find it interesting to note that all models present a similar fit to the confirmed COVID-19 cases data (see Fig. 9 in the Appendix). This makes it impossible to select the best model through visual inspection and comparison of the model fits, or by using model selection methods that do not take the full posterior distributions into account. Although the models might fit the data well, other quantities generated, which are often of interest to the modeller, might be completely incorrect. Moreover, it emphasises the need to test multiple models before any conclusion or inference is undertaken, especially with the complex, hierarchical models.

Although often the Laplace approximation of the normalising constant is sufficient to pick the best model, it was not the case in this epidemiological model selection problem. We can see in Table 1 that the evidence was the highest for the "boundary" models when Laplace approximation was applied. For example, for the sliding window length models, when the Gaussian approximation was applied, the log-evidence was monotonically increasing with the value of $W$ within the range of values that seem reasonable ($W = 1$ to 7). In contrast, with referenced TI, the log-evidence geometry is concave within the range of *a priori* reasonable parameters.

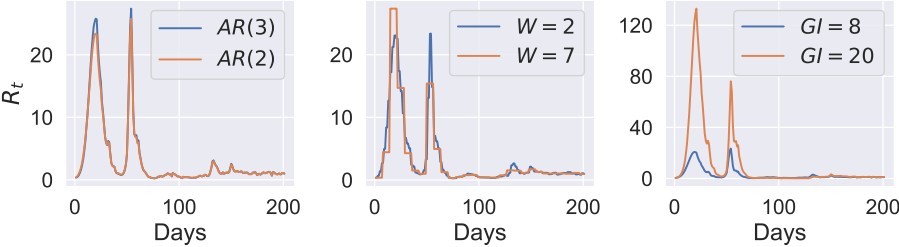

Figure 4: Time-dependent reproduction number generated by models with the highest evidence calculated using the Laplace approximation (orange lines) and referenced TI (blue lines). Note, the fitting data in this example contains superspreading events (which leads to very high values of $R_t$ on certain days) so is not representative of SARS-CoV-2 transmission generally.

## 4 Discussion

The examples shown in Section 3 illustrate the applicability of the referenced TI approach for calculating model evidence. In the *radiata pine* example, referenced TI performed better than the other tested methods in terms of accuracy and speed. When using referenced TI, at $\lambda = 0$ values are sampled from the reference density rather than the prior as in the power posterior method, which should be closer to the original density (in the sense of Kullback–Leibler or Jensen-Shannon divergence). This leads not only to a more accurate estimate of the normalising constant, but also much faster convergence of the MCMC samples. A detailed theoretical characterisation of rates of convergence is beyond the scope of this article, nonetheless the empirical tests presented consistently show faster convergence than with comparative approaches. This is useful especially for evaluating model evidence in complex hierarchical models where each MCMC iteration is computationally demanding.

Although referenced thermodynamic integration and other methods using path-sampling have theoretical asymptotically exact Monte Carlo estimator limits, in practice a number of considerations affect accuracy. For example, biases will be introduced to the referenced TI estimate if one endpoint density substantially differs from another. An example of this and explanation is included in Section A.4.

Furthermore, the discretisation of the coupling parameter path in $\lambda$ can introduce a discretisation bias. For the power posteriors method, Friel et al. (2017) propose an iterative way of selecting the $\lambda$-placements to reduce the discretisation error. Calderhead & Girolami (2009) test multiple $\lambda$-placements for 2- and 20D regression models, and report relative bias for each tested scenario. In the referenced TI algorithm discretisation bias is however negligible — the use of the reference density results in TI expectations that are both small and have low variance, and therefore curvature with respect to $\lambda$. In our framework we use geometric paths with equidistant coupling parameters $\lambda$ between the un-normalised posterior densities, but there are other possible choices of the path constructions, for example a harmonic (Gelman & Meng, 1998) or hypergeometric path (Vitoratou & Ntzoufras, 2017). This optimisation might be worth exploring, however, as illustrated in Fig. 3B, the expectations evaluated vs $\lambda$ are typically near-linear with referenced TI suggesting limited gains, although the extent of this will differ from problem to problem.

In the application to the renewal model for the COVID-19 epidemic in South Korea, we showed that for a complex structured model, hypothesis selection by Laplace approximation of the normalising constant can give misleading results. Using referenced TI, we calculated model evidence for 16 models, which enabled a quick comparison between chosen pairs of competing models. Importantly, the evidence given by the referenced TI was not monotonic with the increase of one of the parameters, which was the case for the Laplace approximation. The referenced TI presented here will similarly be useful in other situations particularly where the high-dimensional posterior distribution is uni-modal but non-Gaussian.

## 5 Conclusions

Normalising constants are fundamental in Bayesian statistics. In this paper we give an account of referenced thermodynamic integration (TI), in terms of theoretical consideration regarding the choice of reference, and show how it can be applied to realistic practical problems. We show how referenced TI allows efficient calculation of a single model's evidence by sampling from geometric paths between the un-normalised density of the model and a judiciously chosen reference density — here, a sampled multivariate normal that can be generated and integrated with ease. Referenced TI method has practical utility for substantially challenging problems of model selection in epidemiology and we suggest it has applicability in other fields of applied machine learning that rely on high-dimensional Bayesian models.

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

# A Appendix

## A.1 Variational Laplace Reference

The conditions to identify an optimal reference normalising constant can be derived by considering a Taylor expansion of the log normalising constant $\log z(\lambda)$ about $\lambda = 0$:

$$\log z(\lambda) \approx \log z(0) + \lambda \, \partial_\lambda \log z(0) + \frac{1}{2}\lambda^2 \, \partial_\lambda^2 \log z(0) \, .$$

The first derivative gives the expectation

$$\partial_\lambda \log z(\lambda) = \mathbb{E}_{q(\lambda;\boldsymbol{\theta})}\left[\log \frac{q(\boldsymbol{\theta})}{q_{\text{ref}}(\boldsymbol{\theta})}\right] \, ,$$

as per the derivation in Equation 2, and the second derivative is a variance

$$
\begin{aligned}
\partial_\lambda^2 \log z(\lambda) \;\; &= \;\; \frac{\int \left(\log \frac{q(\boldsymbol{\theta})}{q_{\text{ref}}(\boldsymbol{\theta})}\right)^2 q(\lambda;\boldsymbol{\theta})d\boldsymbol{\theta}}{\int q(\lambda;\boldsymbol{\theta})d\boldsymbol{\theta}} - \left(\frac{\int \left(\log \frac{q(\boldsymbol{\theta})}{q_{\text{ref}}(\boldsymbol{\theta})}\right) q(\lambda;\boldsymbol{\theta})d\boldsymbol{\theta}}{\int q(\lambda;\boldsymbol{\theta})d\boldsymbol{\theta}}\right)^2 \\
&= \;\; \left\{\mathbb{E}_{q(\lambda;\boldsymbol{\theta})}\left[\left(\log \frac{q(\boldsymbol{\theta})}{q_{\text{ref}}(\boldsymbol{\theta})}\right)^2\right] - \mathbb{E}_{q(\lambda;\boldsymbol{\theta})}\left[\log \frac{q(\boldsymbol{\theta})}{q_{\text{ref}}(\boldsymbol{\theta})}\right]^2\right\} \\
&\geq \;\; 0 \, .
\end{aligned}
$$

As the curvature of $\log z(\lambda)$ is increasing, to first order we see

$$\log z(\lambda) \geq \log z(0) + \lambda \mathbb{E}_{q(0;\boldsymbol{\theta})}\left[\log \frac{q(\boldsymbol{\theta})}{q_0(\boldsymbol{\theta})}\right] \, ,$$

and for the specific case of $\lambda = 1$,

$$\log z \geq \log z_{\text{ref}} + \mathbb{E}_{q_{\text{ref}}(\boldsymbol{\theta})}\left[\log \frac{q(\boldsymbol{\theta})}{q_{\text{ref}}(\boldsymbol{\theta})}\right] \, .$$

This inequality establishes bounds that can be maximised with respect to the position ($\boldsymbol{\mu}$) and scale ($\boldsymbol{S}$) parameters of a reference density such as

$$q_{\text{ref}}(\boldsymbol{\mu}, \boldsymbol{S}; \boldsymbol{\theta}) = q(\boldsymbol{\mu})\exp\left(-\frac{1}{2}(\boldsymbol{\theta} - \boldsymbol{\mu})^{\text{T}}\boldsymbol{S}\,(\boldsymbol{\theta} - \boldsymbol{\mu})\right) \, .$$

Thus the parameters that optimise

$$\max_{\boldsymbol{\mu},\boldsymbol{S}}\left\{\log z_{\text{ref}} + \mathbb{E}_{q_{\text{ref}}(\boldsymbol{\theta})}\left[\log \frac{q(\boldsymbol{\theta})}{q_{\text{ref}}(\boldsymbol{\mu}, \boldsymbol{S}; \boldsymbol{\theta})}\right]\right\} \, ,$$

provide a reference density that is variationally optimal. We note this is an application of the Gibbs-Feynman-Bogoliubov inequality (Bogolubov Jr, 1966; Kuzemsky, 2015; Zhang, 1996), and that finding approximations of this type to the true density is a well-studied problem in machine learning, with well-documented approaches that can be used to determine $q_{\text{ref}}$ variationally (Neal & Hinton, 1998; Jordan et al., 1999). In itself the existence of a variational bound provides no guarantee of being a good approximation to the true normalising constant, and is thus alone not a satisfactory general approach. However as a point of reference from which to estimate the true normalising constant, it provides a first-order optimal density within the family of trial reference functions considered, therefore improving convergence to the MCMC normalising constant in referenced TI.

## A.2 Algorithm

Algorithm 1.

---

**Algorithm 1** Referenced thermodynamic integration algorithm

---

**Input** $q$ - un-normalised density, $q_{\text{ref}}$ - un-normalised reference density, $\Lambda$ - set of coupling parameters $\lambda$, $N$ - number of MCMC iterations

**Output** $z$ - normalising constant of the density $q$

1: Define un-normalised density $q$ and the reference density $q_{\text{ref}}$
2: Calculate $z_{\text{ref}}$ analytically by using the determinant of the covariance matrix, as per Equations 5 or 6 from the main text.
3: **for** $\lambda \in \Lambda$ **do**
4:     Sample N values $\theta_n$ from $q^\lambda q_{\text{ref}}^{1-\lambda}$
5:     **for** $n = 1, 2, \ldots, N$ **do**
6:         Calculate $\log \frac{q(\theta_n)}{q_{\text{ref}}(\theta_n)}$
7:     **end for**
8:     Compute the mean, $\mathbb{E}_\lambda = \frac{1}{N}\Sigma_{n=1}^{N}\log\frac{q(\theta_n)}{q_{\text{ref}}(\theta_n)}$
9: **end for**
10: Interpolate between the consecutive $\mathbb{E}_\lambda$ values to obtain a curve $\partial_\lambda \log(z(\lambda))$
11: Integrate $\partial_\lambda \log(z(\lambda))$ over $\lambda \in [0, 1]$ to get $\log\frac{z}{z_{\text{ref}}}$
12: Calculate $z = z_{\text{ref}} \cdot \exp\left(\log\frac{z}{z_{\text{ref}}}\right)$

---

## A.3 Applications

Here we present additional figures and tables for the examples shown in Section 3 of the paper.

### A.3.1 2D Pedagogical Example with Constrained Parameters

Table 2: Evidence calculated with different methods. Constraint correction refers to imposing the integration limits on the reference as per Equation 8 in the main text. Diagonal covariance means a covariance matrix where only the diagonal (variance) terms are non-zero. Full covariance is a covariance matrix in which all terms can be non-zero. * obtained numerically (Virtanen et al., 2020; Piessens et al., 1983).

| Method | Evidence |
|---|---|
| Exact* | 3.31 |
| Laplace with full covariance | 5.55 |
| Laplace with diagonal covariance + constraint correction | 3.81 |
| Ref TI with full covariance | 4.79 |
| Ref TI with diagonal covariance + constraint correction | 3.33 |

### A.3.2 Benchmarks – *Radiata Pine*

### A.3.3 COVID-19 Model

The COVID-19 model shown is based on the renewal equation derived from the Bellman-Harris process. The details of the model and its derivation are provided in Berah et al. (2021). Here, we give a short overview of the $AR(2)$ model. The model has a Bayesian hierarchical structure and is fitted to the time-series data containing a number of new confirmed COVID-19 cases per day in South Korea from 31-12-2019 to 18-07-2020, obtained from https://opendata.ecdc.europa.eu/covid19/casedistribution/csv. New infections $y(t)$ are modelled by a negative binomial distribution, with a mean parameter in a form of a renewal equation. The number of confirmed cases $y(t)$ is modelled as:

$$y \sim \text{NegBin}(f(t), \phi),$$

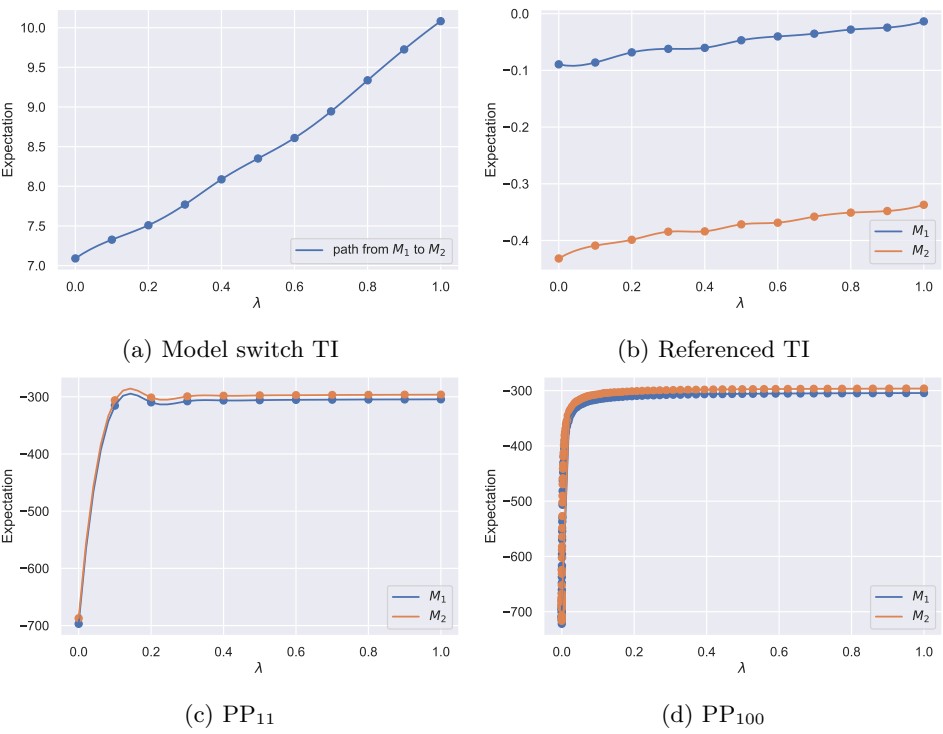

Figure 5: The HMC-evaluated expectation of $\mathbb{E}_{q(\lambda;\theta)}\left[\log\frac{q(\theta)}{q_{\mathrm{ref}}(\theta)}\right]$ vs coupling parameter $\lambda$ is given for models $M_1$ and $M_2$ for four methods of calculating the model evidence: model switch TI (a), referenced TI (b), power posteriors with 11 $\lambda$-placements (c) and power posteriors with 100 $\lambda$ placements (d). Model switch TI (a) creates the path directly between two competing densities, therefore only one line is shown (see Equation 2 in the main text). In each of the plots, the evaluated expectation for a given $\lambda$ is shown with the dot, and the lines connecting the dots were obtained through interpolation.

Table 3: Comparison of Bayes factors for *radiata pine* models for each method. Here we show $BF_{21} = \frac{M_2}{M_1}$ to determine whether model $M_2$ is better than model $M_1$. Both TI and referenced TI methods used 11 equidistant $\lambda$-s. Power posteriors method was used with 11 ($PP_{11}$) and 100 ($PP_{100}$) $\lambda$-s. Third column shows the total number of MCMC steps required to achieve standard error of 0.5%, excluding the warm-up steps. * - using sampled covariance matrix.

| Method | $BF_{21}$ | MCMC steps |
|---|---|---|
| Exact | 4552.35 | - |
| Laplace approximation* | 6309.10 | - |
| Model switch TI | 4557.63 | 2,365 |
| Referenced TI | 4558.71 | 308 |
| $PP_{11}$ | 4463.71 | 41,514 |
| $PP_{100}$ | 4757.82 | 55,000 |

where $\phi$ is an overdispersion or variance parameter and the mean of the negative binomial distribution is denoted as $f(t)$ and represents the daily case data through:

$$f(t) = R_t \sum_{\tau < t} f(t - \tau)g(\tau)\,.$$

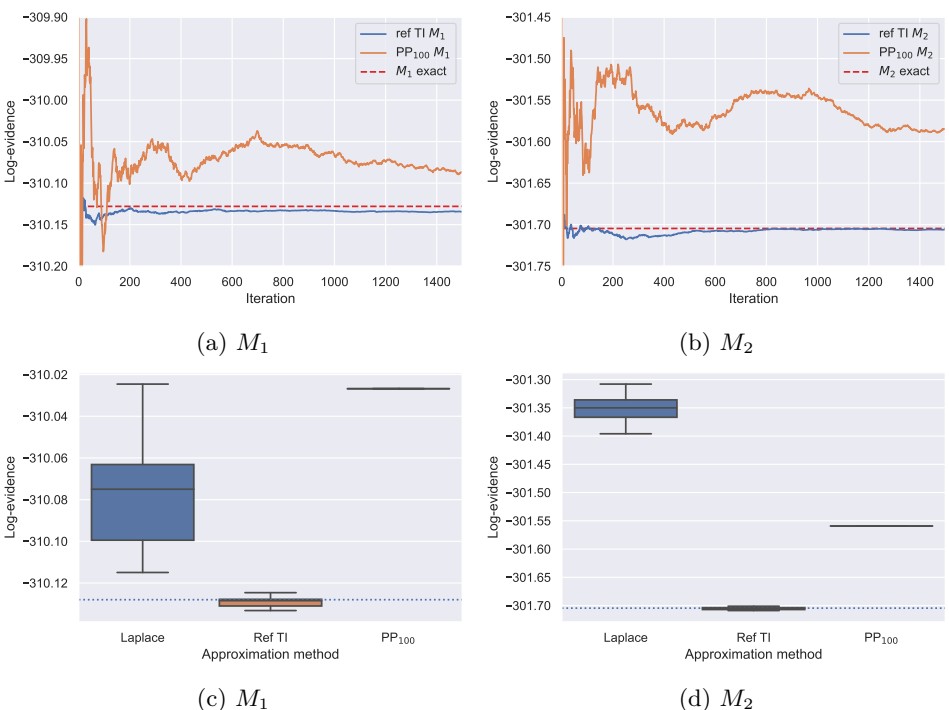

Figure 6: Log-evidence of $M_1$ and $M_2$ for the three algorithms. (a) and (b) show the rolling mean of log-evidence of $M_1$ and $M_2$ over 1500 iterations per $\lambda$ obtained by referenced TI (blue line) and $PP_{100}$ (orange line) methods. The exact value is shown with red dashed line. (c) and (d) show the mean log-evidence of the two models evaluated over 15 runs of the three algorithms. The exact value of the log-evidence is shown with the dotted line.

Here, $g(\tau)$ is a Raleigh-distributed serial interval with mean $GI$, which is discretised as

$$g_s = \int_{s-0.5}^{s+0.5} g(\tau)d\tau \text{ for } s = 2, 3, \dots \text{ and } g_1 = \int_0^{1.5} g(\tau)d\tau\,.$$

$R_t$, the effective reproduction number, is parametrised as $R_t = \exp(\epsilon_t)$, with exponent ensuring positivity. $\epsilon_t$ is an autoregressive process with two-days lag, that is AR(2), with $\epsilon_1 \sim N(-1, 0.1)$, $\epsilon_2 \sim N(-1, \sigma)$ and

$$\epsilon_t \sim N(\rho_1\epsilon_{t-1} + \rho_2\epsilon_{t-2}, \sigma_t) \text{ for } t = \{3, 4, 5, \dots\}.$$

The model's priors are:

$$\sigma \sim N^+(0, 0.2)\,,$$
$$\rho_1 \sim N^+(0.8, 0.05)\,,$$
$$\rho_2 \sim N^+(0.1, 0.05)\,,$$
$$\phi \sim N^+(0, 5)\,,$$
$$GI \sim N^+(0.01, 001)\,.$$

Modification were applied to this basic model, to obtain the different variants of the model as described in Section 3. First group of models analysed was the AR(2) model described above, but with the $GI$ parameter fixed to a certain value instead of inferring that parameter from the data. $AR(3)$ and $AR(4)$ models had additional parameters $\rho_3$ and $\rho_4$, which allow to model the autoregressive process with a longer lag (3- and 4-

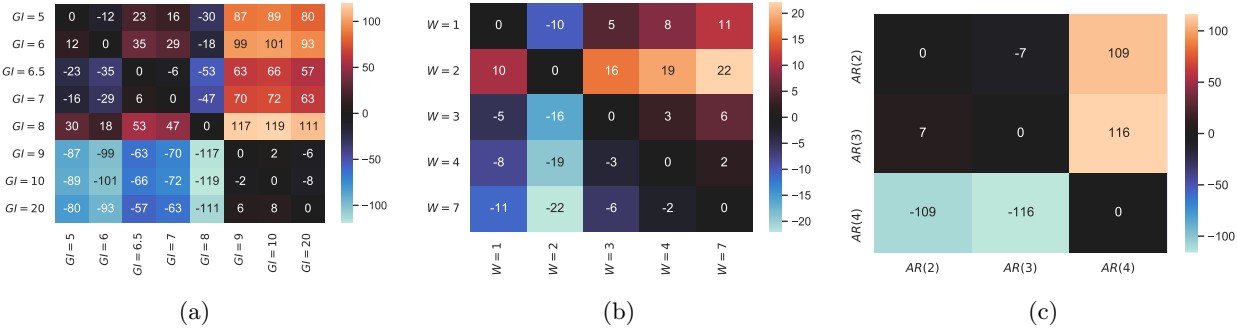

Figure 7: Logarithms of Bayes factors for the analysed COVID-19 renewal models, evaluated using the normalising constants ratios obtained by referenced TI. In each cell, the colour indicates the value of the $BF_{1,2}$ for models $M_1$ (row) and $M_2$ (column). Higher values, that is a brighter orange colour, suggest that $M_1$ is strongly better than $M_2$, and values below 0 in blue palette indicate that $M_1$ is worse than $M_2$. $GI = 8$ performed best out of fixed $GI$ models, $W = 2$ best out of sliding window models, and $AR(3)$ performed better than $AR(2)$ and $AR(4)$. For the interpretation of the BF values see Kass & Raftery (1995).

days respectively). Finally, models $W = k$, $k = 1, .., 7$ were similar to the $AR(2)$ model, but the underlying assumption of these models is that the $R_t$ stays constant for the duration of the length of the sliding window $W = k$.

## A.4 Bias and variance

Although referenced thermodynamic integration and other methods using path-sampling have theoretical asymptotically exact Monte Carlo estimator limits, in practice a number of considerations affect accuracy. For example, biases will be introduced to the referenced TI estimate in practice if one endpoint density substantially differs from another. Then the volume of parameter space that must be explored to produce an unbiased estimate of the expectation cannot be sampled based on the reference density generating proposals within a practical number of iterations. The point is shown for a simple 1D example in Figure 10. Similarly, the larger the mismatch, the higher the variance and slower the expectation is to converge. This illustrates the advantage of using a reference that matches the posterior as closely as possible, as opposed to a typically wide reference like the prior distribution, that gives the characteristic divergence at $\lambda = 0$ with power posteriors. Measures of density similarity in path sampling have been discussed by Lefebvre et al. (2010), however in practical terms there remains much scope for analysis of reference performance in terms of scaling with distribution dimension and type, which should be considered in detail in future work.

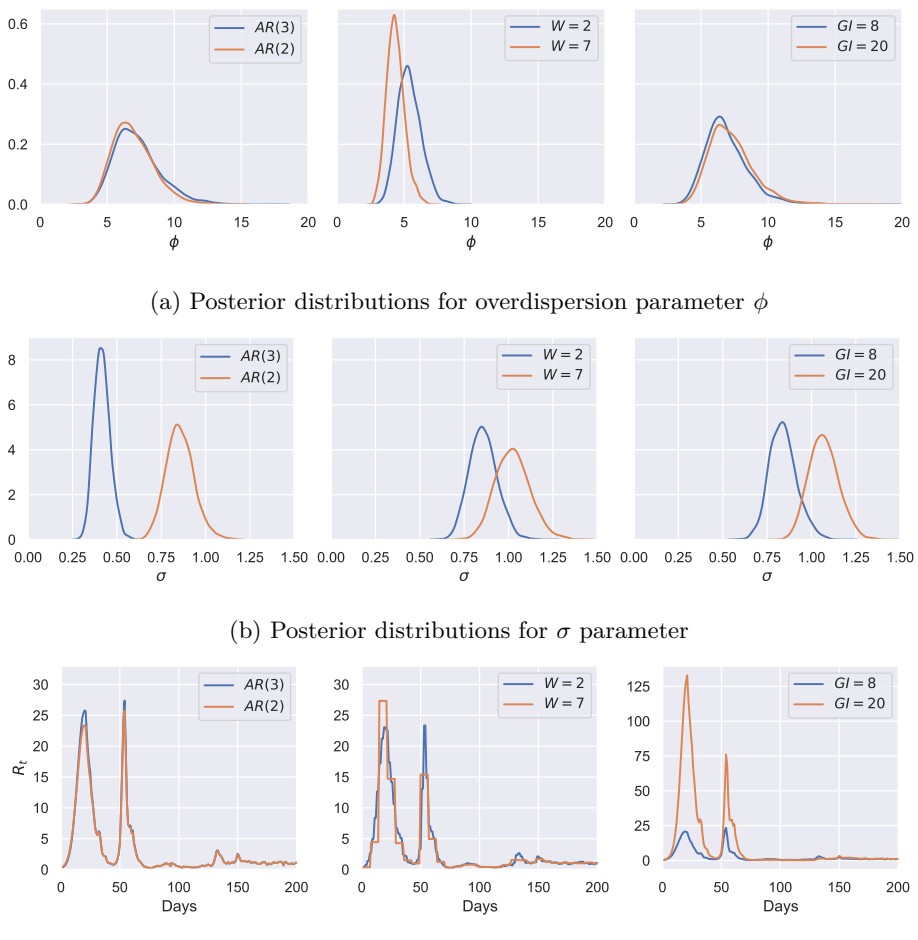

(a) Posterior distributions for overdispersion parameter $\phi$

(b) Posterior distributions for $\sigma$ parameter

(c) $R_t$ generated by the favoured models

Figure 8: Posterior distributions for models' parameters for models favoured by BFs using the Laplace approximation (orange lines) and referenced TI (blue lines).

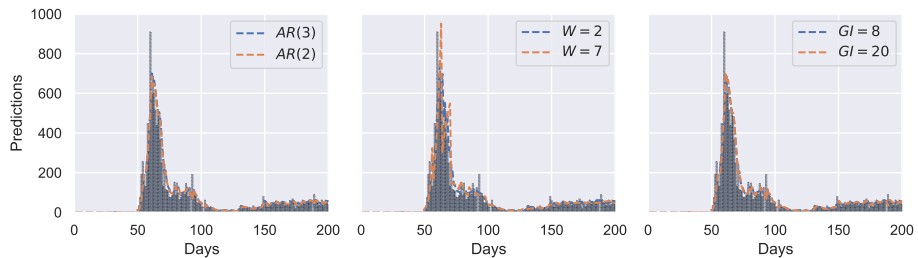

Figure 9: Cases of SARS-CoV-2 infections in South Korea from the data (shown with bars) and the cases predicted by different models. On each graph, predictions made by the model favoured by the Laplace approximation is shown with a blue dashed line, and predictions made by the referenced TI favoured models are shown with an orange dashed line. The lines in all three subplots are largely overlapping, revealing that all models fitted the case data similarly well.

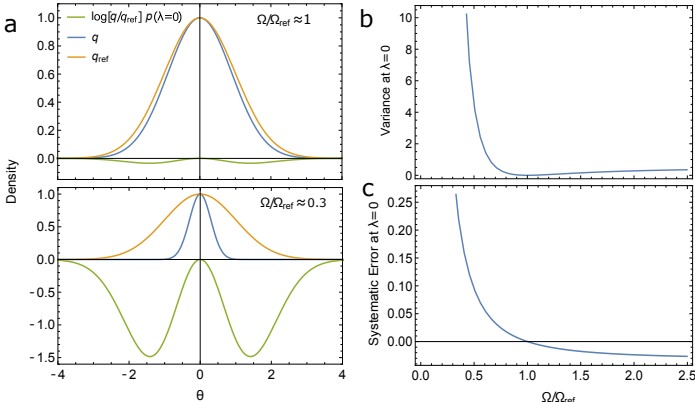

Figure 10: a) 1D examples to illustrate the bias and variance introduced with finite MCMC samples when $q$ and $q_{\mathrm{ref}}$ are mismatched. In these examples $\Omega$ and $\Omega_{\mathrm{ref}}$ denote the domain of the 99% quartiles of $q$ and $q_{\mathrm{ref}}$. b) A mismatch between $q$ and $q_{\mathrm{ref}}$ ($\Omega$ and $\Omega_{\mathrm{ref}}$) causes the variance of $\log\frac{q}{q_{\mathrm{ref}}}$ to increase, requiring more iterations to convergence. c) Similarly the mismatch causes the mass of the distribution for the expectation of $\log\frac{q}{q_{\mathrm{ref}}}$ (evaluated with respect to the reference distribution) to increase beyond the parameter range effectively sampled with finite iterations, in this example corresponding to the 99% quartile of the sampling distribution, thus introducing a bias in the expectation.

