# OpenReview forum: "Application of Referenced Thermodynamic Integration to Bayesian Model Selection"
_TMLR — Rejected by TMLR_

### Review · Reviewer_Md5v · 2022-05-31

**Summary Of Contributions:**

The paper deals with Thermodynamic Integration (TI) to estimate ratios of normalizing constants. The main technical contribution involves the use of variants of the Laplace approximation to build the initial approximating distribution (called reference in the paper). Additionally, the paper presents an interesting application, applying the proposed method to do Bayesian model selection on a task modeling the spread of COVID.

**Broader Impact Concerns:**

No concerns.

**Requested Changes:**

I include all changes that I feel would strengthen the paper. I consider **(R1)** and **(R2)** to be the most important ones, followed by **(R3)**, and finally **(R4)**-**(R5)** (these two final ones are more of a "nice to have thing" which could be interesting to the reader, but not a must).

**(R1)** I think the first contribution mentioned in the introduction could be presented with more details. There's many methods to build flexible and exactly integrable distributions using evaluation of the target's gradient (e.g. normalizing flows). This work presents methods that may be seen as variations of the well-known Laplace approximation. This could be included in the contribution's description (not extensively, but possibly just a few words or a sentence).

**(R2)** More thorough description of related work. See **(W1)**.

**(R3)** An additional task of moderate to large dimensionality. Related to **(W4)**.

**(R4)** An additional baseline for the COVID model, based on the use of the prior as reference (instead of the Laplace approximation). This would show the benefits of using the proposed Laplace-based reference instead of the prior (which, as I understand, is how TI is typically applied).

**(R5)** Analysis of convergence speed using diagonal initial distribution. See **(W2)**.

**Strengths And Weaknesses:**

The paper claims three contributions:

**(C1)** A method to build an initial approximating distribution,

**(C2)** Application to a well-known problem in the statistical literature (with comparisons to other methods),

**(C3)** Application to a Bayesian hierarchical time-series model describing the spread of COVID.

Contributions **(C2)** and **(C3)** are applications. While I consider **(C3)** to be quite interesting, I feel that **(C2)** should be considered a minor contribution, since it is comparing methods for Bayesian model selection over two very simple models (both one-dimensional), using a small dataset (40 samples).

Regarding **(C1)**, I feel that its description in the introduction section could use more details about the method, specifically about the use of Gaussian approximations based on the Laplace method (see "Requested changes" section below).


***Strengths:***

**(S1)** The paper is clearly written (though, as mentioned in the "weaknesses", a thorough related work section would help).

**(S2)** The methods presented are simple to implement. This may be useful to bring attention to this type of methods to people working in areas other than computer science and statistics (e.g. biology, medicine, etc), which may not be familiar with them.

**(S3)** The method shows good results on one relevant and moderately high dimensional problem (modeling COVID data).

**(S4)** While I'm not very familiar with epidemiological models, the main problem tackled, that of COVID spread modeling, appears to be based in a model that's commonly used in the corresponding community.

***Weaknesses:***

**(W1)** A more thorough description of related work would be quite useful, as it would help understanding exactly where this work sits in relation to existing methods, and would help a precise identification of the methodological novel aspects introduced in this work. For instance, section 2.1 explains the use of Laplace approximation to build the initial approximating distribution, and section 2.2 explains the use of a Gaussian approximation whose parameters are estimated using samples obtained via MCMC. While these methods may be considered straightforward (explicitly acknowledged by the authors), it is not completely clear to me whether they have been used/proposed for thermodynamic integration before or not.

**(W2)** The method proposed to deal with target distributions with a limited support (section 3.3) is based on a diagonal (truncated) Gaussian approximation. As acknowledged in the paper, the fact that a diagonal approximation is used affects the approximation quality and the method's convergence speed. I think that an empirical analysis of how is convergence affected would be interesting. (For instance, one could consider a model with unconstrained parameters, and compare the convergence of the estimator using the full initial approximation against the one using the diagonal one.)

**(W3)** The models from sections 3.1, 3.2 and 3.3, while illustrative, all have dimensions 1 or 2, which may not be representative of typical models used in practice.

**(W4)** The method is tested on a single task of moderately large dimensionality.

---

> ### Author Response · Authors · 2022-07-08
> **Response to the Reviewer Md5v**
>
> Strengths:
> (S1) The paper is clearly written...
> (S2) The methods presented are simple to implement....
> (S3) The method shows good results on one relevant and moderately high dimensional problem (modeling COVID data).
> (S4) While I'm not very familiar with epidemiological models....
>
> * We thank the Reviewer for these positive comments about our work.
>
> Weaknesses:
> (W1) A more thorough description of related work would be quite useful, as it would help understand exactly where this work sits in relation to existing methods and would help a precise identification of the methodological novel aspects introduced in this work. For instance, section 2.1 explains the use of Laplace approximation to build the initial approximating distribution, and section 2.2 explains the use of a Gaussian approximation whose parameters are estimated using samples obtained via MCMC. While these methods may be considered straightforward (explicitly acknowledged by the authors), it is not completely clear to me whether they have been used/proposed for thermodynamic integration before or not.
> (W2) The method proposed to deal with target distributions with limited support (section 3.3) is based on a diagonal (truncated) Gaussian approximation. As acknowledged in the paper, the fact that a diagonal approximation is used affects the approximation quality and the method's convergence speed. I think that an empirical analysis of how is convergence affected would be interesting. (For instance, one could consider a model with unconstrained parameters, and compare the convergence of the estimator using the full initial approximation against the one using the diagonal one.)
> (W3) The models from sections 3.1, 3.2 and 3.3, while illustrative, all have dimensions 1 or 2, which may not be representative of typical models used in practice.
> (W4) The method is tested on a single task of moderately large dimensionality.
>
> * We thank the Reviewer for their time spent carefully reviewing our manuscript and appreciate the valuable feedback, which we address in the comments below.
>
> Requested Changes:
> I include all changes that I feel would strengthen the paper. I consider (R1) and (R2) to be the most important ones, followed by (R3), and finally (R4)-(R5) (these two final ones are more of a "nice to have thing" which could be interesting to the reader, but not a must).
>
> (R1) I think the first contribution mentioned in the introduction could be presented with more details. There's many methods to build flexible and exactly integrable distributions using evaluation of the target's gradient (e.g. normalizing flows). This work presents methods that may be seen as variations of the well-known Laplace approximation. This could be included in the contribution's description (not extensively, but possibly just a few words or a sentence).
>
> * Thank you for this suggestion. After discussion and upon thinking about the Reviewers’ comments, we decided to rephrase our contributions. Our main contribution is presenting comprehensive and easy-to-understand examples of calculating model evidence and performing a model selection from the Bayesian workflow perspective, guiding the readers through simple to more complex examples.
> Our primary claim is the approach’s practical applicability to different modelling problems, including real-world high dimensional ones. We will make this clear in the introduction section.
>
> (R2) More thorough description of related work. See (W1).
>
> * Thank you for pointing it out. We will add a more extensive discussion of related methods and expand the introduction section.
>
> (R3) An additional task of moderate to large dimensionality. Related to (W4).
>
> * For this sort of MCMC Bayesian mechanistic models, especially as used in epidemiology, the COVID-19 model presented here is already quite large (up to 200 parameters); this is also the setting where model selection methods like Bayes factors would typically be used. For larger models, e.g. Bayesian Neural Networks, we would not recommend this approach of model selection anyway, as sampling becomes computationally difficult as the dimension of the model increases and consequently geometry of the posterior cannot be sufficiently well approximated with the Gaussian reference.
>
> (R4) An additional baseline for the COVID model, based on the use of the prior as reference (instead of the Laplace approximation). This would show the benefits of using the proposed Laplace-based reference instead of the prior (which, as I understand, is how TI is typically applied).
>
> * Please see the reply to Reviewer 8J6d regarding why we chose not to include more baselines in section 3.4 COVID-19 model.

---

### Review · Reviewer_8J6d · 2022-06-10

**Summary Of Contributions:**

The paper introduces the reader to thermodynamic integration (TI) and examines a referenced TI approach. It explains how reference distributions can be efficiently selected in practical settings. The approach is then demonstrated on multiple problems (toy and benchmark).

**Broader Impact Concerns:**

I do not see broader impact concerns.

**Requested Changes:**

The resolution of the following questions is to me critical for recommendation.:
- Section 3.3 introduces 5 methods in the text, but only 3 of them are shown in the results (in the main text). This should either be changed, or at the very least addressed more clearly/transparently towards the reader.
- Section 3.3: "PP_100 oscillates around it". Where can this be seen? Figure 3a does not suggest oscillation "around" it. Figure 3b even suggests that the PP_100 method converges to the wrong number. Please clarify.
- Section 3.3: The paper states that referenced TI converges fastest. How _precisely_ is "convergence" defined? If this is part of the main appeal of the method and the paper, it should be treated with more detail and care. For example, looking at figure 3 it does not seem like the method converges in 308 MCMC draws; please clarify.
- Section 3.4: Why does this part compare referenced TI only to the Laplace method, not to other suitable sampling-based methods? I am not surprised that "just" doing a Laplace appoximation is not satisfactory. I would expect that all approaches from 3.3. would be applicable here, too, and if they are they might provide more relevant results.
- The plots in the paper are, in my opinion, not suitable for publication. In particular Figure 1 needs changes: Font sizes are inconsistent and too small, fiure height is not consistent, colors / legends are not consistent and thus slightly confusing. Some of this applies to other plots too.

The following is not critical for acceptance, but I believe addressing the issue would strengthen the paper:
- The one-dimensional integral is computed by discretizing along \lambda, interpolating a cubic spline, and then numerically integrating. Why is this a suitable approach? Would it not be preferrable to use an estalished numerical integration scheme? In the discussion it is stated that "the discretization of the coupling parameter path in lambda can introduce a discretisation bias". Would this not be properly addressed by using adaptive integration schemes?

Minor stylistic issues and typos:
- page 1: "the associated hypothesis produced the data" -> "producing" or "which produced"
- Algorithm 1 contains redundancies such as having q, q_ref both as input and first step of the algorithm
- Section A.3.2 exists but has no text. I believe it would be less confusion if there were a small paragraph referencing to figure 5 and table 3; especially since figure 5 contains no information about the fact that it relates to the RP section 3.2.
- Figure 9: The legend describes the colors in a flipped manner: I believe blue is the referenced TI, and orange the Laplace approximation.


**Strengths And Weaknesses:**

_Strengths:_
Overall, the paper is well-written in a way that enables readers unfamiliar with the topic to follow the ideas well. The method of TI is well-introduced, and the two approaches for reference selection are well-presented.

_Weaknesses:_
The claim of favorable performance seems to me not adequately supported.
While Section 3.3 includes the model switch TI and PP_11 in the text, they are not shown in the figures. I would also expect an inclusion of a wider range of methods, e.g. a simple MCMC-sampling baseline, or better a selection of methods from the approaches alluded to in the introduction of the paper ("bridge sampling", "stochastic density of states methods"). Especially with one contribution of the paper being "to bridge the gap from theory and simple examples to application", I expect a more thorough benchmark and comparison to a wider range.
Similarly, Section 3.4 compares the referenced TI approach only to a Laplace approximation; I would again expect a comparison to other sampling-based approaches that would be applicable here (at the very least model switch TI).

With the rather limited novelty of the methodology introduced in the paper, I believe that to be of interest the experimental evaluation of the paper needs to be strengthened.

---

> ### Author Response · Authors · 2022-07-08
> **Response to the Reviewer 8J6d**
>
> Strengths And Weaknesses:
> Strengths: Overall, the paper is well-written...
> Weaknesses: The claim of favorable performance seems to me not adequately supported...
>
> * Firstly, we would like to thank the Reviewer for their time spent on reviewing our manuscript and for the appreciation of our work’s strengths. We are also grateful for receiving this feedback regarding our manuscript’s weaknesses, which we address comment by comment below.
>
> Requested Changes:
> The resolution of the following questions is to me critical for recommendation.:
> Section 3.3 introduces 5 methods in the text, but only 3 of them are shown in the results (in the main text). This should either be changed, or at the very least addressed more clearly/transparently towards the reader.
>
> * Thank you for pointing this out. We will amend the results and Figure 3 to show all 5 methods mentioned in the main text.
>
> Section 3.3: "PP_100 oscillates around it". Where can this be seen? Figure 3a does not suggest oscillation "around" it. Figure 3b even suggests that the PP_100 method converges to the wrong number. Please clarify.
>
> * We agree that the word 'oscillates' is not correct here; the power posterior method is systematically biased and produces an incorrect result in this example. We will change the text to reflect that.
>
> Section 3.3: The paper states that referenced TI converges fastest. How precisely is "convergence" defined? If this is part of the main appeal of the method and the paper, it should be treated with more detail and care. For example, looking at figure 3 it does not seem like the method converges in 308 MCMC draws; please clarify.
>
> * The convergence is defined in this section as “achieving the standard error of 0.5%’'. We will make that clearer to the readers. Following that definition, we get convergence after 308 iterations in Figure 3, because after 308 iterations the fluctuation of the log-evidence estimated by the referenced TI method does not exceed 0.5% . We will make it clearer in the caption of Figure 3.
>
> Section 3.4: Why does this part compare referenced TI only to the Laplace method, not to other suitable sampling-based methods?...
>
> * In section 3.4 we decided to compare our method only with the Laplace approximation for two reasons: firstly, the model switch TI method shown in section 3.3 can only be used to compare pairs of models, which would require calculating the ratio of evidence (7 choose 2) = 21 times; instead, using our referenced TI method we only need to calculate the evidence 7 times. Secondly, we showed in section 3.3 that power posterior method, even with a much larger number of \lambda-s, has systematic issues with estimating the expectation per \lambda around \lambda=0, which impact severly the accuracy of the estimate, as well as it causes significant issues with sampling, which would be even more pronounced for such a complex model. Thirdly, we tried to use a nested sampling method from a readily available package, but since the model is fairly complex and high-dimensional, we failed to obtain any reliable results.
>
> The plots in the paper are, in my opinion, not suitable for publication....
>
> * We agree that Figure 1 is inconsistent, thank you for pointing it out. We will amend Figure 1, taking into account the comments of Reviewer 5hWm as well, and inspect the other figures and make sure they're consistent and clearer.
>
> The following is not critical for acceptance, but I believe addressing the issue would strengthen the paper:
> The one-dimensional integral is computed by discretizing along \lambda, interpolating a cubic spline, and then numerically integrating....
>
> * We agree this should be improved. We will modify evaluation of the 1D integrals to use i) a standard numerical integration method and ii) a model based integration method (Bayesian quadrature). Overall, we expect this not to make a substantial difference. This is because the 1D integrands that ref-TI produces for the final integral are so much flatter and easier to integrate than other methods (e.g. power posteriors), that this step is less important for ref-TI.
>
> Minor stylistic issues and typos:
> page 1: "the associated hypothesis produced the data" -> "producing" or "which produced"
>
> * Thanks for pointing this out, we will amend that.
>
> Algorithm 1 contains redundancies such as having q, q_ref both as input and first step of the algorithm
>
> * Thank you for highlighting this. We will remove the first step from the algorithm.
>
> Section A.3.2 exists but has no text...
>
> * We will add a small paragraph to that Appendix section.
>
> Figure 9: The legend describes the colors in a flipped manner: I believe blue is the referenced TI, and orange the Laplace approximation.
>
> * Thank you for spotting this. There is indeed a mistake in the Figure caption, we will amend that.

---

### Review · Reviewer_5hWm · 2022-06-25

**Summary Of Contributions:**

The paper presents a method for computing the normalizing constant in Bayesian
statistics. The authors' method is based on thermodynamic integration techniques, which
provides means to estimate the ratio between the normalizing constant of two joint
densities via samples. By choosing a reference density whose normalizing constant is known,
we can estimate the normalizing constant of an unknown joint density.
The authors present some experimental results in some synthetic 1 and 2 dimensional
examples, and some real world data sets.


While I found some of the ideas interesting, I do not believe that the authors have
presented a compelling case for the use of the method in practice. There is more work
required in the positioning of the paper, experimental evaluation, and presentation
to warrant publication.



**Requested Changes:**

See above.

**Strengths And Weaknesses:**

Related Work:
The paper lacks a related work section where the authors position their work in relation
to prior work. In its current form, the authors have discussed prior work in an ad-hoc
manner around Section 2 with the method.
- Here, the authors should also discuss in detail differences with other work which uses
  references and why TI may be expected to do better.


Experiments:
- My main concern here is that the authors haven't evaluated their method
  against other methods for estimating the posterior, such as Bayesian quadrature and the
  sampling based methods highlighted in Section 1.
- It would be nice to also look at time for convergence instead of just number of
  iterations. It is not clear to me that each iteration would be computationally more
  efficient than other methods.
- Why are some figures/tables in the appendix? It seems like they all belong in the
  experiments section.
- The authors should do a better job of explaining the results to the reader. For
  instance, what does success look like in Fig. 5. How do I know that the authors' method
  does better than other methods. These things need to be explained in the appendix.


Other:
- In Section 2.2, if you can generate samples, can't you directly estimate the normalizing
  constant? What advantages does using it to construct a reference density first offer?
- Consider using "Experiments" or "Evaluation" as an alternative title for Section 3.


Presentation/clarity:
- It might be useful to be explicit in the notation about the data (say X) and write
  qi(\theta, X), pi(\theta|X) etc around equation 1. I eventually figured out this is
  what the authors meant,  but was left confused initially.
- Fig 1A is referenced in the intro, but doesn't appear until much later. It might be
  useful to illustrate these ideas with a more intuitive figure early on.
- p(\lambda;\theta), q(\lambda;\theta) are not defined prior to equation 2. Are these
  quantities viewed as densities on lambda parametrized by theta or the other way around.
  The notation suggests the former, but the derivation in equation 2 suggests the latter.
- Explain how the reader should think about z1, z2 after equation 2 (i.e. z2 will be the
  normalizing constant of a reference density while z1 will be the target). This will
  prime the reader for what's coming, and also not leave anyone confused as to the
  significance of equation 2.
- "the main desirable features are that it should be easily formed without special
   consideration or adjustments and that zref should be analytically integratable and
   account for as much of z as possible". Here it is not clear what "easily formed",
   "without special consideration or adjustment" or "account for as much of z" mean.
- Figure 1 A could be clearer. Consider using densities that are more far apart so that
  the transition from one to another is more apparent.
- Figure 1 caption: explain what model evidence is for a reader who may not be familiar.

---

> ### Author Response · Authors · 2022-07-08
> **Response to the Reviewer 5hWm**
>
> * We thank the Reviewer for taking time to read our manuscript and provide valuable feedback. Below we addressed all comments and we will make appropriate changes in the text.
>
> Strengths And Weaknesses:
> Related Work: The paper lacks a related work section....
>
> * We agree with the Reviewer that a separate and more focused writing around related work is required to position our approach and contributions. We will add a detailed paragraph on related methods and position our contributions better in the background section. Please also see our reply to Reviewer Md5v, section R1.
>
> My main concern here is that the authors haven't evaluated their method against other methods for estimating the posterior....
>
> * We first thank the reviewer for raising this concern and apologise for not making it clear in text the reasons behind this omission. A major contribution of this paper is making available and accessible a methodology that can be used with little changes for  Bayesian model selection. To our knowledge, most of the methods for calculating normalising constants are not available in the form of off-the-shelf packages in Python. The only off-the shelf package which we have found is one for nested sampling.  Bayesian quadrature using standard Gaussian process approaches are not suitable for very high dimensional integration problems.
> We will certainly add the comparison with the Nested sampling approach in section 3.3 along with adding the comparison with bayesian quadrature for 1-D example.
>
> It would be nice to also look at time for convergence instead of just number of iterations....
>
> * Thank you for that suggestion. We will measure and add the CPU time required for running the model to the text.
>
> Why are some figures/tables in the appendix?...
>
> * We decided to put some figures and tables in the appendix due to the 12 pages constraint of the paper. We are happy to move them into the main text if the Reviewers think that's more appropriate.
>
> The authors should do a better job of explaining the results to the reader....
>
> * Thank you for raising this issue. In Figure 5, the ideal situation is if the curve constructed by expectations per \lambda is smooth and easy to integrate (such as for the model switch TI and ref TI). We will certainly add that explanation to the figure caption. We will also inspect other figures and tables and add similar clarification where necessary to help the reader understand the presented outcomes.
>
> In Section 2.2, if you can generate samples, can't you directly estimate the normalizing constant....
>
> * Using probabilistic programming languages such as Stan or NumPyro, we can easily sample from an arbitrary density function, even un-normalised, without knowing its normalising constant. However, just being able to sample from it does not give us the normalising constant directly. We need methods such as the referenced TI method presented here to be able to calculate it.
>
> Consider using "Experiments" or "Evaluation" as an alternative title for Section 3.
>
> * Thank you for this suggestion. We feel that ‘Applications’ is an appropriate title for section 3, as it directly relates to the title of the manuscript, and in our opinion highlights its main purpose, i.e. practical applicability of the method.
>
> It might be useful to be explicit in the notation about the data (say X) and write qi(\theta, X), pi(\theta|X) etc around equation 1....
>
> * Thank you for pointing this out. We will amend the notation in equation 1 as suggested.
>
> Fig 1A is referenced in the intro, but doesn't appear until much later....
>
> * Thank you for this idea, we will add a more intuitive figure earlier in the text.
>
> p(\lambda;\theta), q(\lambda;\theta) are not defined prior to equation 2. Are these quantities viewed as densities on lambda parametrized by theta or the other way around. The notation suggests the former, but the derivation in equation 2 suggests the latter.
>
> * q(\lambda;\theta) is defined above equation 2 on page 2. We will clarify what p(\lambda;\theta) is as well in the text. Both of these quantities are densities of \theta parametrised by the path coordinate \lambda.
>
> Explain how the reader should think about z1, z2 after equation 2....
>
> * Thank you for this suggestion. We will add a sentence about it after Equation 2.
>
> "the main desirable features are that it should be easily formed without special consideration or adjustments and that zref should be analytically integratable and account for as much of z as possible". Here it is not clear what "easily formed", "without special consideration or adjustment" or "account for as much of z" mean.
>
> * Thank you for pointing that out. We will add clarification to this text.
>
> Figure 1 A could be clearer. Consider....
> Figure 1 caption: explain what model evidence is....
>
> * Thank you for these two suggestions. We will make Figure 1 clearer and add explanation of the model evidence in the caption. *

---

### Decision · Action_Editors · 2022-08-09

**Recommendation:** Reject

**Comment:**

While the reviewers found the proposed method and the ideas behind it interesting, they believe that the authors have not presented a compelling case for its use. The work has not been positioned well in relation to the existing methods. A related work section in which other approaches to the problem are discussed with their pros and cons, and in relation to the proposed method is missing. The main concern is the lack of proper empirical comparison with methods for estimating the posterior. Due to the empirical nature of the work, comparing against a wider range of baselines seems to be necessary. The reviewers also believe that the writing of the paper needs to be significantly improved. This includes better explanation of the related work and the results, and highlighting the main contributions. Given all the above, I believe the work in its current form is not ready for publication. It requires a major revision and a new round of evaluation. The reviewers have listed the changes/improvements they would like to see in the paper. I strongly recommend that the authors take them into account when they prepare the new version of their work.